Succession of bacterial communities on carrion is independent of vertebrate scavengers

Dangerfield Cody R. 1
Frehner Ethan H. 1
Buechley Evan R. 1 2
Şekercioğlu Çağan H. 1 3
Brazelton William J. william.brazelton@utah.edu 1
1 School of Biological Sciences, University of Utah , Salt Lake City , UT , USA
2 HawkWatch International , Salt Lake City , UT , USA
3 Department of Molecular Biology and Genetics, Ko University , Istanbul , Turkey
Curiel Yuste Jorge
Electronic publication date: 2020 Jun 10
Publication date: 2020
Volume: 8
Electronic Location ID: e9307
Received 2019 Aug 28; Accepted 2020 May 17
Copyright: ©2020 Dangerfield et al.
Copyright year: 2020
Copyright holder: Dangerfield et al.
License: This is an open access article distributed under the terms of the Creative Commons Attribution License, which permits unrestricted use, distribution, reproduction and adaptation in any medium and for any purpose provided that it is properly attributed. For attribution, the original author(s), title, publication source (PeerJ) and either DOI or URL of the article must be cited.
License URL: https://creativecommons.org/licenses/by/4.0/

Keywords: Succession, Carrion, Decomposition, Bacterial diversity, Forensic microbiology

Funding: National Science Foundation University of Utah Undergraduate Research Opportunities Program This work was funded by the National Science Foundation in the form of a Graduate Research Fellowship (ERB) and by the University of Utah through start-up funds and the Undergraduate Research Opportunities Program. The funders had no role in study design, data collection and analysis, decision to publish, or preparation of the manuscript.

==============================
The decomposition of carrion is carried out by a suite of macro- and micro-organisms who interact with each other in a variety of ecological contexts. The ultimate result of carrion decomposition is the recycling of carbon and nutrients from the carrion back into the ecosystem. Exploring these ecological interactions among animals and microbes is a critical aspect of understanding the nutrient cycling of an ecosystem. Here we investigate the potential impacts that vertebrate scavenging may have on the microbial community of carrion. In this study, we placed seven juvenile domestic cow carcasses in the Grassy Mountain region of Utah, USA and collected tissue samples at periodic intervals. Using high-depth environmental sequencing of the 16S rRNA gene and camera trap data, we documented the microbial community shifts associated with decomposition and with vertebrate scavenger visitation. The remarkable scarcity of animals at our study site enabled us to examine natural carrion decomposition in the near absence of animal scavengers. Our results indicate that the microbial communities of carcasses that experienced large amounts of scavenging activity were not significantly different than those carcasses that observed very little scavenging activity. Rather, the microbial community shifts reflected changes in the stage of decomposition similar to other studies documenting the successional changes of carrion microbial communities. Our study suggests that microbial community succession on carrion follows consistent patterns that are largely unaffected by vertebrate scavenging.

Introduction

Carrion, or dead animal tissue, provides a nutrient-rich resource for a wide array of organisms. At the smallest scale, both geographically and in organisms affected, carrion contributes nutrients to soils via nutrient leaching, thereby affecting microbial communities in soil near the carcass (Howard, Duos & Watson-Horzelski, 2010; Parkinson et al., 2009; Parmenter & MacMahon, 2009). Scaling upwards, carrion can be seen more directly as a food source to the many necrophagous arthropods and vertebrate scavengers (Jordan et al., 2015). Furthermore, larger-scale impacts of carrion have been well documented in the massive die-off of salmon and cicada, which lead to large increases in resources and nutrient availability that affect a myriad of organisms including microbes, plants, fungi, and vertebrates (Hocking & Reynolds, 2011; Hocking & Reynolds, 2012; Jordan et al., 2015; Tiegs et al., 2009; Tiegs et al., 2011; Yang, 2004). The versatile methods by which carrion can be produced and consumed gives it the potential to impact many facets of an ecosystem, and the pathway by which it decomposes depends on the environmental conditions and the interactions that form among the organisms that compete over its resources.

The decomposition of carrion typically follows a consistent progression that is often categorized into stages based on the descriptions in Payne (1965): fresh, bloat, active decay, advanced decay, and putrid dry remains. A body enters the fresh stage immediately after death, and depletion of internal oxygen triggers autolysis of the cells. Shortly after death, endogenous microbes begin to metabolize the body and produce volatile compounds. As the activity of these microbes fills the body cavity with gases, causing the carcass to distend, the carcass transitions into the bloat stage. Active decay follows the bloat stage when the body cavity ruptures, releasing the gases and allowing invertebrates to consume a larger portion of the soft tissue within the body cavity (Carter, Yellowlees & Tibbett, 2007). After the removal of most of the soft tissue, invertebrate activity decreases, and the carcass transitions into the advanced decay stage. The carrion enters the putrid dry remains stage when the carcass has desiccated, leaving only bones and small amounts of skin and hair (Goff, 2009; Payne, 1965).

The high nutrient content of carrion makes it a highly sought-after resource for many organisms (Hanski, 1987; Janzen, 1977; Wilson & Wolkovich, 2011). Due to the high level of competition, the organisms that consume carrion have developed behaviors in order to monopolize the nutrients of carcass for themselves. These complex interactions among microbes and scavenging fauna, along with abiotic factors (e.g., precipitation and temperature), often impact the duration and occurrence of the previously described stages (Carter, Yellowlees & Tibbett, 2008; Carter, Yellowlees & Tibbett, 2010; Comstock et al., 2015; Galloway, Jones & Parks, 1989; Payne, 1965; Rozen, Engelmoer & Smiseth, 2008; Shukla et al., 2017). For example, some microbes that begin to metabolize the carrion soft tissue after an animal’s death also produce toxins in order to hinder consumption from other organisms (Blandford et al., 2019; Burkepile et al., 2006; Janzen, 1977). In response, some scavengers, such as the turkey vulture (Cathartes aura), have developed an unusually high tolerance to decomposer-produced toxins (e.g., botulinum toxin), and the highly acidic conditions present in their hindgut reduce the likelihood of microbes surviving consumption and infecting the vulture itself (Beasley, Olson & DeVault, 2015; DeVault Jr, Rhodes & Shivik, 2003; Roggenbuck et al., 2014). Furthermore, behaviors such as the burial of carcasses have developed in both vertebrate and invertebrate scavengers in order to seclude the carrion from climatic conditions, microbes, and other scavengers to slow decomposition and secure the resources for themselves (Frehner et al., 2017; Rozen, Engelmoer & Smiseth, 2008; Shukla et al., 2017). Burying beetles (Nicrophorus spp.) further suppress competition with microbes by excreting antimicrobial exudates on the carcass (Barnes, Gennard & Dixon, 2010; Rozen, Engelmoer & Smiseth, 2008; Shukla et al., 2017). In doing so, these beetles limit decomposition and can delay the carcasses from entering the bloat or active decay stages (Shukla et al., 2017), which are mainly dictated by microbial and insect activity (Finley, Eric Benbow & Javan, 2015; Goff, 2009; Payne, 1965).

Researching these interactions is important to understand how carrion decomposition impacts nutrient cycling and the importance that carrion has on ecosystems (Barton, 2015; Barton et al., 2013). There are also forensic benefits to understanding these interactions as the pattern of succession on carrion and cadavers has the potential to determine postmortem time intervals (PMI; Amendt et al., 2011; Anderson, 2015). Historically, a majority of studies focused on forensic entomology to determine PMI Byrd & Allen, 2001; Michaud & Moreau, 2009; Payne, 1965; Schoenly & Reid, 1987). However, recent studies have used DNA sequencing technology to investigate the potential for these microbes to serve as PMI indicators (Burcham et al., 2016; Finley, Eric Benbow & Javan, 2015; Guo et al., 2016; Metcalf et al., 2013; Pechal et al., 2014) and further characterize the carrion microbiome (Hyde et al., 2013; Pechal et al., 2013; Pechal et al., 2018; Pechal et al., 2019; Weatherbee, Pechal & Eric Benbow, 2017). These studies have further characterized the microbial communities associated with carrion decomposition and how seasonal changes and macroinvertebrates impact those microbial communities.

Research into the interactions between animal scavengers and microbes have typically focused on scavenger avoidance of microbe-laden carrion (Blandford et al., 2019; Burkepile et al., 2006). It has long been suggested that scavengers may facilitate the spread of pathogens throughout an ecosystem (Houston & Cooper, 1975), but scavengers’ involvement in disease ecology is complex and requires further study (Beasley, Olson & DeVault, 2015). A recent study has suggested that macroinvertebrates feeding on carrion may act as vectors of microbes among carcasses (Pechal et al., 2019), but to our knowledge, no study has investigated the influence of vertebrate scavengers, or the lack thereof, on the microbial communities of carrion in a terrestrial environment. In this study, we used environmental DNA sequencing and vertebrate scavenging data to investigate decomposition dynamics and potential impacts that vertebrate scavengers have on the microbiome of carrion in the Great Basin Desert of Utah. Specifically, we tested the hypothesis that vertebrate scavengers disrupt the typical progression of carcass decomposition and introduce variability into the succession of microbial communities on carrion.

Figure 1 Photographs of the decomposition progression at one of the seven sites.

The photos from this site were selected for clarity purposes, and all other sites had similar carcass composition.

Methods

Study sites and field data collection

In this study, we investigate the bacterial communities of bovine carcasses in the Grassy Mountain region of Utah, USA (40.87°N, -113.03°W) from May to June, 2015. To do so, we experimentally placed juvenile domestic cow (Bos taurus) carcasses (n = 7) in the study site and monitored their decomposition using camera traps to identify vertebrate scavenger activity and by collecting tissue samples at regular intervals to identify the progression of microbial communities. We collected tissue samples from each of the carcasses during five sampling periods (Day 1, Day 4, Day 12, Day 18, and Day 26) (Fig. 1). The calves were obtained from one local Utah dairy and had died from natural causes either during or shortly after birth. The carcasses were collected on the day of birth/death, and were kept frozen for 1–6 months until their placement in the field. Carcasses thawed during transport through the desert but were still cold at the time of final placement and of the first sample collection (i.e., Day 1). They were placed at least 3 km apart and fixed to a concealed stake in the ground to prevent scavengers from removing the complete carcass. All carcasses were placed and sampled on the same day. The carcasses weighed between 18.6 and 26.9 kg. Carcasses were placed on sites that included sparse Utah juniper (Juniperus osteosperma), greasewood (Sarcobatus vermiculatus), and widely distributed cheatgrass (Bromus tectorum). The soil in the study area is composed of loose to moderately compacted limnological sediments, including gravels and clays. The study area is arid and largely homogenous. Study area temperatures varied between 7−40 °C, and there was no precipitation during the experiment. Field methods were approved by the University of Utah Institutional Animal Care and Use Committee (Protocol Number 15-06001) and by the US Bureau of Land Management through a joint letter of agreement.

The carcasses were equipped with Bushnell Trophy Cam HD motion-activated cameras to monitor vertebrate scavenging activity. The cameras were programmed to take 1 photo when triggered, with a 10-s delay between subsequent photos to reduce saturation of photos from the same animal visitation event. All photos collected over the course of the study were entered into CameraBase Version 1.7 (Tobler, 2007), a camera-trap photo management platform in Microsoft Access. We analyzed each of these photos individually and identified any vertebrates present in the photos to species. We identified arrival times after carcass placement and duration of presence at carcass for each scavenger species.

Tissue samples (∼15 cm3 in size) were excised from the hind leg of each carcass. Samples contained muscle, skin, fat, and hair during the early stages of decomposition, and skin and hair in the latter stages of decomposition when no soft tissue remained. Soil samples directly adjacent to where the carcass was placed were acquired during the first sampling period for two of the seven sites. Soil samples for the remaining sites were taken 5 m from the carcass during the second sampling period due to concerns that the earlier soil samples were collected in too close proximity to the carcass where they may have been contaminated by the carcass itself. The microbial diversity results indicated that all soil samples were very similar to each other, alleviating this concern. DNA was extracted from all carcass and soil samples using the PowerSoil DNA Isolation Kit (MO BIO Laboratories, Carlsbad, CA, USA) according to the manufacturer’s instructions and stored at −20  °C.

Bacterial 16S rRNA gene sequencing

The samples were submitted to the Michigan State University genomics core facility for bacterial 16S rRNA gene amplicon sequencing. The V4 region of the 16S rRNA gene (defined by primers 515F/806R) was amplified with dual-indexed Illumina fusion primers as described by Kozich et al. (2013). Amplicon concentrations were normalized and pooled using an Invitrogen SequalPrep DNA Normalization Plate. After library QC and quantitation, the pool was loaded on an Illumina MiSeq v2 flow cell and sequenced using a standard 500 cycle reagent kit. Base calling was performed by Illumina Real-Time Analysis (RTA) software v1.18.54. Output of RTA was demultiplexed and converted to fastq files using Illumina Bcl2fastq v1.8.4. Paired-end sequences were filtered and merged with each other (to form, on average, 253 bp merged sequences) with USEARCH 8 (Edgar, 2010), and additional quality filtering was conducted with the mothur software platform (Schloss et al., 2009) to remove any sequences with ambiguous bases and more than 8 homopolymers. Chimeras were removed with mothur’s implementation of UCHIME (Edgar et al., 2011). The sequences were pre-clustered with the mothur command pre.cluster (diffs =1), which reduced the number of unique sequences from 1,136,609 to 784,953. This pre-clustering step removes rare sequences most likely created by sequencing errors (Schloss & Westcott, 2011).

Bacterial diversity analyses

The unique, pre-clustered sequences were considered to be the operational taxonomic units (OTUs) for this study and formed the basis of all alpha and beta diversity analyses, as in our previous study (Dangerfield, Nadkarni & Brazelton, 2017). Sequence reads were not rarefied for alpha diversity and evenness calculations because there was no correlation between diversity indices and sequencing depth for this study (Table S1). Taxonomic classification of all sequences was performed with mothur using the SILVA reference alignment (SSURefv123) and taxonomy outline (Pruesse, Peplies & Oliver Glöckner, 2012). Taxonomic counts generated by mothur and edgeR were visualized using the R package phyloseq 1.20.0 (McMurdie & Holmes, 2013).

Statistical analyses

Alpha diversity and evenness were calculated with the Shannon, invsimpson, and simpsoneven calculators provided in the mothur package (Schloss et al., 2009). Differences between alpha diversities and evenness were tested for significance using the Dunnett-Tukey-Kramer test, which accounts for multiple comparisons among samples with unequal sizes and variances (Lau, 2013). Beta diversity was measured using the Morista-Horn biodiversity index, as implemented in mothur. This index was chosen because it reflects differences in the abundances of shared OTUs without being skewed by unequal numbers of sequences among samples. Differences between community compositions were tested for significance using AMOVA (analysis of molecular variance) as implemented in mothur (Pruesse, Peplies & Oliver Glöckner, 2012). Morisita-Horn community dissimilarity among samples was visualized using a nonmetric multidimensional scaling (NMDS) plot. This plot was generated using the ordinate and plot ordination commands in phyloseq (McMurdie & Holmes, 2013). The ggplot2 function stat_ellipse was added to draw 95% confidence level ellipses (assuming t-distribution) in the NMDS plot (Wickham, 2016). Environmental variables (temperature, scavenger counts, and scavenging duration) were fitted to the community composition ordination with the envfit function in the vegan package (Oksanen et al., 2019). Differences in the relative abundance of OTUs between stages was measured using the R package edgeR (Robinson, McCarthy & Smyth, 2009) as recommended by McMurdie & Holmes (2014). The differential abundance of an OTU (as measured in units of log2 fold change by edgeR) was considered to be statistically significant if it passed a false discovery rate threshold of 0.05. Taxa were determined characteristic to that specific stage if they were found differentially abundant in one stage compared to all other stages and soil. These taxa are referred to as “characteristic taxa” for the purposes of this paper. To investigate potential environmental contamination of carcass samples, OTUs with at least 20 sequence counts among all samples were assigned to either fresh carcass or soil using the sink-source Bayesian approach of SourceTracker2 v2.0.1 (Knights et al., 2011) with rarefying to 66,001 sequences for sinks and 16,828 sequences for sources. Similar results were obtained without rarefying sequence counts (Figs. S1, S2). The one carcass sample that was determined to be contaminated by soil via SourceTracker2 was excluded from alpha and beta diversity analyses.

Data Availability

All sequence data are publicly available at the NCBI Sequence Read Archive under BioProject PRJNA525153.

Results

Carcass decomposition

Sampling periods were categorized into stages of decomposition based on physical interpretation of the carcasses (Fig. 1) as determined from camera trap photographs taken of each carcass and as described by Payne (1965). Day 1 was determined to be “fresh”, as the carcasses were kept frozen promptly after death. The carcasses was determined to be in the “bloat” stage at Day 4, as evidenced by the body cavity becoming distended by gases emitted during microbial decomposition. The bloat stage may have begun earlier than Day 4, but was not apparent in the camera trap photographs. The later sampling periods (Day 12, 18, and 26) were all categorized as the “active decay” stage, because of the large decrease in carcass size and the presence of skin tissue on the carcass.

Vertebrate scavengers that fed at the carcasses included American badger (Taxida taxus), common raven (Corvus corax), coyote (Canis latrans), kit fox (Vulpes macrotis), turkey vulture, and white-tailed antelope squirrel (Ammospermophilus leucurus). Turkey vultures were the most frequent scavenger to feed at the carcasses, and the majority of vertebrate scavenging occurred between Day 4 and Day 12 of the study (Table 1).

Table 1 Remarkably little scavenging activity observed at cow carcass sites.

Summary of scavenging activity per species per site, including number of individuals and minutes of scavenging activity observed.

Site	Species	Individuals	Scavenging Duration (Min)	
Site 1	Coyote	1	3	
Turkey Vulture	1	9	
Total	2	12	
Site 2	American badger	1	1	
Total	1	1	
Site 3	Common Raven	1	4	
Coyote	1	1	
White-tailed Antelope Squirrel	6	22	
Total	8	27	
Site 4	Common Raven	3	15	
Kit Fox	3	7	
Turkey Vulture	9	172	
Total	15	194	
Site 5	Coyote	2	6	
White-tailed Antelope Squirrel	1	1	
Total	3	7	
Site 6	White-tailed Antelope Squirrel	1	3	
Total	1	3	
Site 7	Common Raven	16	101	
Coyote	1	1	
Turkey Vulture	4	85	
Total	21	187	

Impact of scavenging on bacterial communities

To identify the impact that scavenging had on bacterial community composition, we compared Morisita-Horn dissimilarities between high- and low-scavenging sites directly after the peak of scavenging. Sites 4 and 7 were considered to be the “high scavenging” sites as they experienced 88% of the total scavenging duration observed during the study (Table 1). Differences in bacterial community composition between low-scavenging and high-scavenging sites were minimal and not significant (AMOVA Fs = 0.603245, p = 0.636). Additionally, fitting of scavenging parameters (individuals per week and scavenging duration per week) to the NMDS ordination (Fig. 2) yielded no significant correlations.

Figure 2 Nonmetric multidimensional scaling (NMDS) plot showing bacteria community shifts associated with the stages of decomposition.

The ellipses indicate where 95% of samples within a sample period are expected to occur on the plot.

To identify individual operational taxonomic units (OTUs) that scavengers may have introduced to the carcasses, we contrasted the relative abundances of OTUs in high-scavenging sites to low-scavenging sites directly after the major scavenging events (Day 12 samples). This comparison yielded 39 OTUs that were differentially abundant in the two high-scavenging sites in comparison to low-scavenging sites. Only a few OTUs were present in both high-scavenging sites.

We also examined the relative abundance patterns of OTUs classified as genera reported by previous studies to be associated with macroinvertebrates (Singh et al., 2018; Dharne et al., 2008; Gupta et al., 2011; Gupta et al., 2014; Lee et al., 2014; Shukla et al., 2017; Tóth et al., 2008; Weatherbee, Pechal & Eric Benbow, 2017). All macroinvertebrate-associated genera reached peak relative abundances during the later sampling periods except for Providencia and Myroides (Fig. 3).

Figure 3 Abundance of genera associated with macroinvertebrates.

All genera except Myroides and Providencia exhibited abundance increases in the latter sampling periods.

Bacterial community changes over time

Microbial community differences are visualized in Fig. 2, where each data point represents the overall bacterial community composition of one sample and the distance between points represents the dissimilarity between samples. There are two primary shifts in community composition: one from Day 1 to Day 4 and a second from Day 4 to later sampling periods (Fig. 2). The 95% confidence ellipses show consistent separation between these sampling periods. The only exception to this pattern was a single sample from Day 4 that clusters with soil samples in Fig. 2. We examined the bacterial community composition of this outlier sample in more detail with SourceTracker2 (Knights et al., 2011), which revealed that 83% of OTUs in the outlier sample could be confidently assigned to soil. Therefore, we concluded that this sample had been contaminated with soil during sampling and/or handling, and we excluded this sample from all downstream analyses.

Figure 2 also shows that the bacterial community composition of all carcasses at later sampling dates (Days 12, 18, 26) are highly similar and not significantly different from each other. The patterns visualized in the NMDS ordination were tested with an AMOVA that confirmed significant differences between the Day 1, Day 4, and later sampling periods (Table 2). These three clusters of bacterial community composition (Day 1, Day 4, and Days 12-18-26) correspond to the three stages of decomposition identified by physical interpretation of the carcasses (Day 1 = “fresh”, Day 4 = “bloat”, and Days 12-18-26 = “active decay”).

Table 2 Microbial community compositions of cow carcasses are distinct at different stages of decomposition and are distinct from that of soil.

Description	Comparison	Fs	P-value	
Overall	All 6 sample groups	9.30736	<0.001*	
	Day 1 vs. Day 4	5.88847	0.001*	
	Day 4 vs. Day 12	7.56111	<0.001*	
	Day 12 vs. Day 18	0.406973	0.738	
	Day 18 vs. Day 26	−0.22461	0.978	
	Soil vs. Day 1	11.173	0.001*	
	Soil vs. Day 4	7.47528	<0.001*	
	Soil vs. Day 12	14.5593	<0.001*	
	Soil vs. Day 18	16.8612	0.002*	
	Soil vs. Day 26	16.8429	<0.001*	
Notes.

Statistics from AMOVA analysis of Morisita-Horn dissimilarities are shown.

* significant.

Proteobacteria was the most common phylum in the fresh stage, accounting for 48% of microbial community composition, decreasing to 11% for both the bloat and active decay stages (Fig. 4). Conversely, Firmicutes abundance increased as decomposition progressed. Firmicutes increased from 31% in the fresh stage to 72% and 84% of the microbial community in bloat and active decay, respectively. Moraxellaceae represented 30% of the total abundance of the fresh stage (Fig. 4), whereas Moraxellaceae only accounted for 2% and 0.8% of the total abundance in bloat stage and active decay, respectively. By contrast, Clostridia was dominant in the bloat stage, accounting for 70% of the total abundance, and accounting for only 3% of total abundance in the fresh stage (Supplemental Information).

Figure 4 Bacterial communities of each stage of decomposition.

Taxa that are present throughout all stages or in soil is represented by “Ubiquitous taxa”. The remain taxa are uniquely abundant within stage of decomposition. Uniquely abundant taxa that had <1% abundance was grouped into “Low abundance taxa”.

Taxa characteristic to each stage of decomposition

The taxonomic classifications (at order and family level) of all OTUs identified as characteristic to each stage of decomposition (as defined by differential abundance analysis described in Methods) are shown in Fig. 4. Each bar in Fig. 4 represents the total community composition of each stage, in which “Ubiquitous taxa” represents taxa that are equally abundant across two or more groups and the remaining taxa in each bar represent OTUs that are characteristic to that stage. There were 323 OTUs that were characteristic to the fresh stage, and these taxa accounted for ∼62% of the total abundance in that stage. These fresh stage taxa are dominated by Moraxellaceae, Flavobacteriaceae, Pseudoalteromonadaceae, Planococcaceae, Staphylococcaceae, and an unclassified Bacillales family. The bloat stage contained fewer characteristic taxa (106 OTUs) than the fresh stage, and these taxa accounted for ∼56% of the total abundance in that stage. The characteristic bloat taxa mainly consisted of Clostridia OTUs. In particular, five Clostridia OTUs accounted for ∼49% of the bloat stage’s total abundance. The active decay stage contained 230 characteristic OTUs that accounted for ∼64 percent of the total abundance. Active decay still contained OTUs from Clostridia, but these OTUs are different than those Clostridia OTUs in the bloat stage and account for less of the total abundance. Active decay stage contained other prominent characteristic taxa from Flavobacteriaceae, Enterococcaceae, Xanthomonadales and an unclassified Bacillales family. All characteristic OTUs, their proportional abundances, and taxonomic classifications are reported in the Supplemental Information.

Alpha diversity

Figure 5 reports the OTU Shannon diversity index of each stage of decomposition. Alpha diversity was highest during the fresh stage (Shannon diversity range = 4.8–7.2) and decreased during the bloat stage (Shannon diversity range = 3.3–4.4). Diversity was more variable but greater on average in the active decay stage (Shannon diversity range = 3.8–6.7). The shifts in Shannon diversity between fresh and bloat stages and between bloat and active decay stages passed a Dunnett-Tukey-Kramer significance test. Similar patterns were also evident with the Inverse Simpson and Evenness (from Simpson) indices.

Figure 5 Average Shannon alpha diversity and standard error between decomposition stages and soil.

Significance of each comparison was conducted by the Dunnett-Tukey-Kramer test. All comparisons were significantly different from one another except fresh vs active decay. ∗ = significant (p < 0.05).

Discussion

No correlation between vertebrate scavenging and bacterial community composition

Carrion is a valuable resource for which many organisms compete, including vertebrate scavengers, macroinvertebrates, and microbes. Although a few studies have documented the association of bacteria with macroinvertebrates on carrion (Pechal et al., 2013; Pechal et al., 2019; Rozen, Engelmoer & Smiseth, 2008; Shukla et al., 2017), this is the first study to investigate the impact of vertebrate scavengers on the bacterial community of carrion in a terrestrial environment. Our study site in an arid region of Utah was well-suited for this experiment because of its unusually sparse vertebrate scavenging activity (EH Frehner, ER Buechley, H Şekercioğlu, unpublished data) Studying the succession of bacterial communities in the absence of vertebrate impacts would have been difficult or impossible in a typical environment without significant experimental manipulations.

One potential consequence of vertebrate scavenging could be the disruption of the progression of carcass decomposition, introducing variability into the succession of microbial communities. Our results did not provide any support for this hypothesis. No impact of scavenging activity on the bacterial community composition of carrion was observed. Overall, the bacterial communities of carcasses that experienced vigorous vertebrate scavenging (e.g., 15 individuals and 3.2 hours of activity) were highly similar to the bacterial communities of carcasses at the same stage of decomposition but with almost no vertebrate scavenging (e.g., 1 individual and 1 minute of activity).

Subtle effects of vertebrate scavenging may not be evident in the overall bacterial community composition, which was strongly correlated with the stage of decomposition (Figs. 2 and 4). Therefore, we tested whether any of the 784,953 bacterial operational taxonomic units (OTUs) in this study significantly shifted in abundance after the peak in scavenging activity on Day 12. Surprisingly, only 39 of these OTUs exhibited significant differential abundance between high-scavenging and low-scavenging sites, highlighting the remarkable similarity among all carcasses, especially at the active decay stage. None of these 39 OTUs were particularly abundant in both high-scavenging sites. If the impact of scavenging on microbial composition is rapid and transient, it may not have been captured by our sampling density of once every few days, so future studies should consider sampling at much more frequent time points (Benbow & Pechal, 2017).

Although we saw no evidence of bacterial taxa consistently associated with vertebrate scavenging, we did find taxa that have been previously associated with turkey vultures (Roggenbuck et al., 2014). However, the abundances of these taxa were not higher in sites that experienced more scavenging by turkey vultures. Furthermore, Roggenbuck et al. (2014) speculated that these taxa are most likely derived not from the turkey vultures, but from the carrion they consume.

Bacterial taxa associated with macroinvertebrates

Although our study was not designed to investigate the impact of macroinvertebrates, we observed several genera of microbes that have been previously associated with scavenging macroinvertebrates (Dharne et al., 2008; Gupta et al., 2011; Gupta et al., 2014; Lee et al., 2014; Shukla et al., 2017; Tóth et al., 2008; Weatherbee, Pechal & Eric Benbow, 2017). The abundance pattern of most of the taxa are consistent with other studies (Singh et al., 2018; Shukla et al., 2017; Weatherbee, Pechal & Eric Benbow, 2017). With the exception of Providencia and Myriodes, most of these genera exhibited increased abundance when the carcasses entered into active decay. This increase in macroinvertebrate-associated taxa is consistent with the increases of macroinvertebrate activity that typically occur during the active decay stage (Carter, Yellowlees & Tibbett, 2007; Finley, Eric Benbow & Javan, 2015; Matuszewski et al., 2010; Payne, 1965). Unfortunately, we were not able to include observations of macroinvertebrates in our study design.

Stages of decomposition have consistent bacterial communities

During this study, the carcasses exhibited three stages of decomposition (fresh, bloat, and active decay), and each stage was characterized by a unique community of bacteria. Although the weather, geography, and vertebrate scavenging activity of our study was notably different than in previous studies, our observed patterns of bacterial community succession are remarkably similar to those previously reported for carrion (Hyde et al., 2013; Pascual et al., 2017; Pechal et al., 2013; Pechal et al., 2014; Pechal et al., 2018; Weatherbee, Pechal & Eric Benbow, 2017). Pechal et al. (2013) reported that Proteobacteria were dominant in the early stages of decomposition and declined as decomposition progressed and that Firmicutes abundances progressively increased in the later stages of decomposition. Similarly, 8 of the 14 families that exhibited distinct temporal patterns during the decomposition process in Pascual et al. (2017) exhibited similar patterns in our study (Fig. 4).

Our results are also consistent with those of previous studies that have shown consistent aerobic/anaerobic microbial community shifts during the progression of decay (Burcham et al., 2016; Singh et al., 2018; Finley, Eric Benbow & Javan, 2015; Goff, 2009; Hyde et al., 2013; Metcalf et al., 2013; Pascual et al., 2017; Pechal et al., 2013; Pechal et al., 2014). Abundance patterns of OTUs from Moraxellaceae, all of which are known to be aerobic (Pascual et al., 2017), exemplify this pattern: they represent 30.3% of total abundance in the fresh stage, 2.4% in the bloat stage (when anaerobic Clostridia and Enterobacteriaceae are dominant), and 0.78% in the active decay stage. Furthermore, the dominance of anaerobic bacteria during the bloat stage is exemplified with Clostridia comprising 70% of the microbial community in the bloat stage (Dataset S1). This dominance of a few taxa during the bloat stage is associated with low alpha diversity values (Fig. 5), whereas Pascual et al. (2017) observed the highest alpha diversity during the bloat stage (described as the putrefaction stage) in forest ecosystems.

Most other studies observe more rapid progression through the stages of decomposition (Pascual et al., 2017; Pechal et al., 2013; Pechal et al., 2014). It is possible that carcasses during the last sampling period of our study were in “advanced decay”, but there was no discernible change in tissue composition of the carcasses compared to the previous two sampling periods. The delay in decomposition is most likely the result of climatic factors. Previous studies investigating carrion microbial communities were conducted in areas with much higher humidity, whereas this study experienced arid conditions and high temperatures during a period with no precipitation. Active decay has been shown to be limited or hindered by hot and dry climates similar to the conditions present in this study, resulting in partial mummification (Galloway, Jones & Parks, 1989). During this process, carcasses develop a mummified shell over the skeleton as the skin desiccates while macroinvertebrate activity continues underneath, in the body cavity. This partial mummification may explain the prolonged intact composition of the carcass in the latter sampling periods (Fig. 1) associated with an increased abundance of macroinvertebrate-associated bacterial taxa (Fig. 3).

Conclusion

In this study, we investigated microbial succession associated with decomposition of cow carcasses that experienced notably little vertebrate scavenging activity. Our results provided no support for the hypothesis that isolated, intense vertebrate scavenging events affect the progression of carcass decomposition or the bacterial communities on carrion. Instead, the bacterial community composition of all carcasses consistently reflected the stage of decomposition, regardless of vertebrate scavenging activity. A more expansive study with additional time points and additional replicate carcasses may have been able to detect more subtle shifts in the abundance of individual taxa in response to scavenging within each stage of decomposition.

Our results are remarkably similar to those of other studies conducted in wetter, milder conditions with greater vertebrate scavenging activity (Burcham et al., 2016; Hyde et al., 2013; Pascual et al., 2017; Pechal et al., 2013; Pechal et al., 2014; Pechal et al., 2019; Weatherbee, Pechal & Eric Benbow, 2017), suggesting that bacterial community succession on carrion follows consistent patterns that are largely unaffected by many external factors. However, we observed differences in the timing of decomposition, most likely due to the arid climate, and perhaps associated with this, differences in the trajectory of microbial diversity through later stages of decomposition (Fig. 5) compared with studies in wetter environments (e.g., Pascual et al., 2017).

Our results provide additional support for the use of microbial community composition as a reliable forensic indicator of the timing of carcass decomposition. In particular, this study was able to show that arid environmental conditions and variable activity of vertebrate scavengers did not greatly affect the overall succession of microbial communities on cow carcasses.

Supplemental Information

Dataset S1 Additional data showing the distributions of OTUs and taxonomic classifications among all samples

Interactive Krona plots, and the full count table showing the distribution of each OTU among all samples.

Click here for additional data file.

Table S1 No correlation between Shannon diversity and sequence counts

(r2 = 9.402e − 06, p = 0.985).

Click here for additional data file.

Figure S1 Neither soil nor the first sampling time point of the cow carcasses were major sources of microbial taxa to later sampling time points

These results reflect a SourceTracker2 analysis without rarefaction.

Click here for additional data file.

Figure S2 Neither soil nor the first sampling time point of the cow carcasses were major sources of microbial taxa to later sampling time points

These results reflect a SourceTracker2 analysis after rarefaction down to the smallest dataset.

Click here for additional data file.

Additional Information and Declarations

Competing Interests

Author Contributions

Animal Ethics

Field Study Permissions

DNA Deposition

Data Availability

Evan Buechley is employed by HawkWatch International. The authors declare there are no competing interests.

Cody R. Dangerfield analyzed the data, prepared figures and/or tables, authored or reviewed drafts of the paper, and approved the final draft.

Ethan H. Frehner performed the experiments, analyzed the data, prepared figures and/or tables, authored or reviewed drafts of the paper, and approved the final draft.

Evan R. Buechley conceived and designed the experiments, performed the experiments, analyzed the data, prepared figures and/or tables, authored or reviewed drafts of the paper, and approved the final draft.

Çağan H. Şekercioğlu conceived and designed the experiments, authored or reviewed drafts of the paper, and approved the final draft.

William J. Brazelton conceived and designed the experiments, analyzed the data, authored or reviewed drafts of the paper, and approved the final draft.

The following information was supplied relating to ethical approvals (i.e., approving body and any reference numbers):

Field methods were approved by the University of Utah Institutional Animal Care and Use Committee (IACUC Protocol Number 15-06001).

The following information was supplied relating to field study approvals (i.e., approving body and any reference numbers):

Field work was approved by the US Bureau of Land Management through a joint letter of agreement.

The following information was supplied regarding the deposition of DNA sequences:

All sequence data are publicly available at the NCBI Sequence Read Archive: PRJNA525153.

The following information was supplied regarding data availability:

Data is available at GitHub: https://github.com/Brazelton-Lab.

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
