# Peer review of "Succession of bacterial communities on carrion is independent of vertebrate scavengers"

_PeerJ, doi:10.7717/peerj.9307_

## Round 0.1 · original submission · Major Revisions

Dear authors

There are some major concerns raised by the two independent reviewers that should be addressed. There is a serious problem with clarity of the text and edition, which needs to be solved. Also one of the reviewers complains that you have not incorporated recent bibliography to the citations and raised some concerns on the experimental design. Both were very critical with the discussion, which should be substantially improved according to their concerns. These and the rest of the comments should be properly addressed in a future revised version.

Reviewer 1 ·

Basic reporting

Some small editorial comments: authors' use the word "utilize" incorrectly (ex line 93; utilize = a strategy is put to practical advantage or a chemical or nutrient is being taken up and used effectively; see https://www.quickanddirtytips.com/education/grammar/use-versus-utilize?page=1); there are double periods at the ends of some sentences while others are lacking punctuation (ex lines 46 and 101); wording is ambiguous, vague, or sometimes wrong maybe to avoid sounding too much like author's original wording? (ex lines 51-54); common names are not capitalized unless they are a proper noun (ex line 83; lines 211-215)

I think the paper is at some times insufficient in it's literature review, lacking citations of recent research in the field and leaving out citations entirely (ex lines 47-50; lines 51-62; lines 70-75; line 93-96; lines 337-346).

Experimental design

There doesn't appear to be any control or discussion of a control: Were any carcasses used a s a control? (Such as caged as to exclude as much vertebrate scavengers as possible?) If not, was a baseline of microbial diversity established at t=0? so like at the time of placement? That could also serve as a control of sorts. Or a soil sample? I think this should be addressed at least.

Lines 226-230: this method alone is insufficient to determine if the scavenger activity was the reason for lack of diversity. See comment about controls.

Can't make the conclusion in line 381-382. You didn't test any other factor but scavenging. Insects could be homogenizing the microbial community....

Validity of the findings

Data analyzed appropriately, analysis is recreateable, methods robust, figures appropriate, figure captions appropriate.

Some speculation is made but not tempered as such:

Lines 226-230: this method alone is insufficient to determine if the scavenger activity was the reason for lack of diversity. See comment about controls.

Can't make the conclusion in line 381-382. You didn't test any other factor but scavenging. Insects could be homogenizing the microbial community....

Reviewer 2 ·

Basic reporting

The concept of identifying the microbes in combination with vertebrate scavenging is excellent. However, the lack of clarity in this manuscript made it cumbersome to read and difficult to understand the logic and justification of the experimental design. Understandable, carrion research is often difficult to perform due to logistic and financial constraints, but the lack of replication in this manuscript is a source of concern.

Major Concerns:

• I am concerned about the sampling regime to make broad statements about the results for the microbial portion. It is under-described, ambiguous, and not well justified and thus a major weak point of the manuscript. It is unclear if specimens form only two of the seven carcasses were collected at each of the five days? where they different carcasses each of the five days? Please add more details to clarify if the samples carcasses were sampled each day, if not, why?

• There is an entire section the “bacterial taxa associated with macroinvertebrates” but nowhere in the methods is there a section described about how inverts were collected, processed, identified, etc. The assumption that they are insect based without collecting and identifying bacteria from the insect is a weak section of the manuscript.

• How did the temperature relate to the decomposition? Four days for fresh is a long time, unless the carcasses were not thawed prior to field deposition and the ambient temperature was closer to 4C.

• The authors describe the bacterial community composition at broad scales (phyla, order, family), with 16S data why not delve into pattern differences at the genus or OTU/ESV levels?

• In the discussion, the authors highlight the novelty of excluding vertebrate scavengers in their study design, but many of the other carrion-microbe studies focus on the insect activity and have anti-vertebrate scavenging cages placed on top of the carcasses to reduce scavenging pressures from vertebrates (works by Weiss et al, Metcalf et al., Hyde et al., Pechal et al., Weatherbee et al., etc.).

• The comment about vertebrates “could be important vectors” bacteria is an interesting one (and one in which I agree about potential for bacterial movement), but there is no data to support the potential for this type of activity in the current manuscript. 16S is detecting DNA and not viability, and there did not appear to be any analyses concerning actual transfer from carcass to vertebrate even based on a reduction of bacterial taxa post-scavenging events.

Experimental design

See above section.

Validity of the findings

See above section.

Additional comments

The authors should review the manuscript carefully, and check for typos, spacing errors, and punctuation errors. Below are more specific comments (minor concerns) that the authors may want to consider addressing:

Line 51: Payne had six stages in terrestrial systems
Line 71: Provide a scale for “quickly” and are you referring to exogenous or endogenous microbes?
Line 77: Please expand on “harsh conditions present in their hindgut”.
Line 80: change “has” to “have”
Line 83: “Burying Beetles” is not capitalized, and “spp.” is not italicized.
Line 84: Need citations to support the sentence.
Line 99-100: Citations after “Scavengers act as a vector of
100 dispersal for many microbes…”
Line 101: End of sentence needs only one period.
Line 101: Beginning sentence, is this true across all ecosystems (e.g., aquatic and terrestrial)?
Line 103: Is this really the use of “eDNA” in the classical sense?
Line 112: Justification behind the regular sampling intervals? What were they?
Line 115: Were the carcasses thawed prior to field deposition? How long were carcasses kept frozen? We know that long-term freezing will impact the microbial communities. Also, thawing bodies will impact the microbial succession (Pechal et al 2018 JFS) of human bodies, and thus one may assume would also impact the successional patterns of other vertebrates.
Line 116: were all carcasses placed into the field on a single day?
Line 123: I would recommend moving this information closer to line 112; also, please justify why these intervals were chosen.
Line 136: Why the muscle tissue? How far were the biopsies made? Was a punch used or a standardized volume? Was the depth of muscle tissue sampled for the microbes? Again, how was this standardized?
Line 136: why not all carcasses? Also, were these soil samples taken prior to carcass placement? What does “Directly adjacent mean” in terms of distance from the carcass? Why not sample from all the carcasses each time?
Line 138: Again, it is confusing about the sampling regime given that tissue samples were only
Line 142: What was the length of the amplicons?
Line 152: Remove the second parenthesis after the citation.
Line 161: Are the data presented for the lack of correlation? If not, it would be great to see in the supplemental material.
Line 193: Again, the results from the comparisons would be a good addition to the supplemental material.
Line 194: if samples were only collected from two carcasses during each day, and one carcass had to be removed, how many sampling days have an n of 1?
Line 222: Data/results/figures to support no statistical difference?
Line 228: Remove “using edgeR”, as it is in the methods.
Line 233: How were “macroinvertebrate-associated genera” identified? What host species were used? Did this overlap with the insects found during decomposition? How do you know if they are insect related if no insects were collected?
Line 237: Unnecessary line.
Line 299: “this is the first…” may want to add in a terrestrial system given Burkepile’s work. Also, a new study by Blandford et al. 2019 that you may want mentioned in the discussion.

---

## Round 0.2 · Major Revisions

I decided to return you the MS for major revisions based on the comments of the independent reviewer. Besides the grammar issues raised by the reviewer, there are some methodological flaws that has not been properly addressed in the corrected version, e.g. the soil sampling in the methods is still insufficiently explained. It is also not clear how they have adequate replicates to present those results, based on the experimental design. The author´s should, therefore, justify better their experimental design and support their findings using more robust statistics. The authors should also consider discussing how their sample size impacted their results. Please read carefully reviewer´s comments and address them in a revised version

As a personal criticism, I see that it is an article that does not present clear hypotheses, and this affects the scientific quality of the work. I suggest that the authors also develop hypotheses based on the state of the art and the objectives they are trying to meet. The author´s also need to contextualize better why is so important their science, what kind of information and what relevance does it have to understand the interaction between vertebrates scavengers and microbiota in carrion decomposition.

Reviewer 2 ·

Basic reporting

Where will the analytical code be deposited?

Experimental design

Please further clarify the justification for the soil sampling in the main text.

Validity of the findings

No comment.

Additional comments

Review the document for plural agreement and word choice.

Review document for use and appropriateness of adverbs.

Payne 1965 stages were not properly addressed in the revisions (Line 51)

“Postmortem time intervals” is not abbreviated “PMI” (Line 160).

Review line 236 for clarity.

Despite the comment that “Soil sampling was added during the study and was not executed entirely consistently across all carcasses, as explained in the text”. Where in the revision, is the fact explained to readers that are not familiar with experimental design/justification? One cannot assume the justification from the details provided. Please further clarify in line 283 why the soils were only collected adjacent (<1 m) from 2 of the 7 carcasses on day one, and then at 5 m the remaining sites during day 4.

My apologies for not inquiring in the previous version, but is the code for all analyses going to be made publicly available, if so where?

In the alpha-diversity results section, please provide quantities for how much diversity increased/decreased.

In line 400, I would recommend being more forthright about the fact that these 39 OTUs were only detected in a single carcass (as indicated in the discussion).

Line 59, due to the data collected in this study, you can perform a post-hoc power analyses to determine how many carcasses would be needed to detected differences. Further, in the discussion section about no consistent patterns of bacterial community change after scavenging, there is not discussion about the potential temporal aspects of when sampling occurred relative to the scavenging event. This factor too could be important in identifying transient scavenger derived taxa.

Finally, I do not believe there is an adequate discussion about the sample size of the study and the power/effect size of their results. Carrion work in general can be difficult to obtain sufficient independent, replicates. The authors may consider discussing how their sample size impacted their results.

---

## Round 0.3 · accepted · Accept

Author´s have properly addressed the minor concerns of the reviewer, highlighting the limitations of this data set in detecting potential subtle differences in microbial composition due to vertebrate scavenger activity and the necessity of more expansive studies to properly test the hypotheses. Still, results obtained in the study are of great interest, so I am glad to finally recommend this manuscript for publication in PeerJ.